# First Insights on Resistance and Virulence Potential of *Escherichia coli* from Captive Birds of Prey in Portugal

**DOI:** 10.3390/antibiotics13050379

**Published:** 2024-04-23

**Authors:** Rita Magalhães, Raquel Abreu, Gonçalo Pereira, Eva Cunha, Elisabete Silva, Luís Tavares, Lélia Chambel, Manuela Oliveira

**Affiliations:** 1CIISA—Centre for Interdisciplinary Research in Animal Health, Faculty of Veterinary Medicine, University of Lisbon, 1300-477 Lisbon, Portugal; rita-magalhaes@edu.ulisboa.pt (R.M.); rmsilva@fmv.ulisboa.pt (R.A.); goncalopereira@fmv.ulisboa.pt (G.P.); elisabetesilva@fmv.ulisboa.pt (E.S.); ltavares@fmv.ulisboa.pt (L.T.); moliveira@fmv.ulisboa.pt (M.O.); 2AL4AnimalS—Associate Laboratory for Animal and Veterinary Sciences, 1300-477 Lisbon, Portugal; 3BioISI—Biosystems and Integrative Sciences Institute, Faculty of Sciences, University of Lisbon, 1749-016 Lisbon, Portugal; lmchambel@ciencias.ulisboa.pt; 4cE3c—Centre for Ecology, Evolution and Environmental Changes and CHANGE—Global Change and Sustainability Institute, Faculty of Sciences, University of Lisbon, 1749-016 Lisbon, Portugal

**Keywords:** falconry, one health, virulence, antimicrobial resistance, ERIC-PCR, *Escherichia coli*

## Abstract

Captive birds of prey are often used for pest control in urban areas, while also participating in falconry exhibitions. Traveling across the country, these birds may represent a public health concern as they can host pathogenic and zoonotic agents and share the same environment as humans and synanthropic species. In this work, *Escherichia coli* from the cloacal samples of 27 captive birds of prey were characterized to determine their pathogenic potential. Isolates were clustered through ERIC-PCR fingerprinting, and the phylogenetic groups were assessed using a quadruplex PCR method. Their virulence and resistance profile against nine antibiotics were determined, as well as the isolates’ ability to produce extended-spectrum β-lactamases (ESBLs). The 84 original isolates were grouped into 33 clonal types, and it was observed that more than half of the studied isolates belonged to groups D and B2. Most isolates presented gelatinase activity (88%), almost half were able to produce biofilm (45%), and some were able to produce α-hemolysin (18%). The isolates presented high resistance rates towards piperacillin (42%), tetracycline (33%), and doxycycline (30%), and 6% of the isolates were able to produce ESBLs. The results confirm the importance of these birds as reservoirs of virulence and resistance determinants that can be disseminated between wildlife and humans, stressing the need for more studies focusing on these animals.

## 1. Introduction

The One Health concept was established to address planet sustainability issues from multiple perspectives, aiming to achieve a balance between human, animal, and environmental health [1,2]. Since antimicrobial compounds are crucial for safeguarding both human and animal health, the development of antimicrobial resistance constitutes a serious worldwide problem and is being considered a critical global threat by the World Health Organization as it is expected to be responsible for the death of ten million people by 2050. Accordingly, a One Health perspective must be implemented to manage this global problem [3].

As wildlife has become more adapted to anthropogenic environments and widespread in urban areas, some species, such as pigeons and gulls, are frequently regarded as resident birds in the metropolitan areas of Europe, and this has several negative impacts. For instance, these birds may pose substantial health risks, having already been described as responsible for disease transmission to humans [4]. The occurrence of multidrug-resistant bacteria in these synanthropic species has been frequently described in recent years [5,6], with the detection of resistance strains in gulls having increased in tandem with the growth of the human population [7].

Antimicrobial resistance in enteric microbiota can be used as an indicator of resistance dispersion in the environment, allowing us to infer on the extent of selective pressure, due to the use of antimicrobials in humans and animals [8]. Commensal intestinal bacteria, such as *Escherichia coli*, have been described as good fecal indicators [9]; moreover, this bacterial species includes relevant multidrug-resistant strains, including carbapenem-resistant and extended-spectrum β-lactamases-producing strains, with known resistance to fluoroquinolones, aminoglycosides, and trimethoprim–sulfamethoxazole [10]. Apart from including commensal strains, enteric *E. coli* can be categorized into distinct pathotypes according to their pathogenicity and virulence traits, which include enterohemorrhagic (EHEC), enteropathogenic (EPEC), enterotoxigenic (ETEC), enteroinvasive (EIEC), enteroaggregative (EAEC), diffusely adherent (DAEC), and adherent-invasive (AIEC) *E. coli* [11,12]. Together, these traits make *E. coli* a relevant model for investigating resistance transmission pathways within ecosystems [9]. Simultaneously, studies on virulence traits of *E. coli* from synanthropic species revealed the presence of pathogenic isolates from phylogenetic groups B2 and D in these birds, significantly associated with the ability to produce capsule and yersiniabactin; moreover, isolates from phylogenetic group B2 were described as relevant carriers of genes associated with ESBL resistance [13]. A gene that encodes intimin, an outer membrane protein, was also described in *E. coli* from three different species of synanthropic birds from Spain, proving their potential as vectors for the dissemination of virulence determinants [14].

In an attempt to manage the increase in unwelcome wildlife in urban areas, integrated falconry programs have been successfully established, becoming an alternative to other pest control strategies that can pose ethical concerns, such as the use of pyrotechnics [15]. The same birds are often used for display purposes in environment conservation programs, as they can easily captivate the attention of the target audiences and evoke an emotional connection [16]. This means that, during their activities, these animals contact not only with synanthropic birds, particularly seagulls and pigeons, but also with humans, which unveils an underlying potential public health risk resulting from interactions with these animals.

This study aimed to characterize the virulence and pathogenic potential of *E. coli* isolates from captive birds of prey working in contact with humans and synanthropic species. The data obtained are expected to contribute to understanding the role of these birds as potential carriers and environmental spreaders of bacteria with important resistance and virulence traits as key information for safeguarding One Health, and to confirm the importance of implementing preventive measures to counter the spread of this problem.

## 2. Results

### 2.1. Sampled Animals and Isolate Identification

Animals sampled for this study belong to 10 distinct bird species (Table 1), with *Parabuteo unicinctus* being the most frequent species sampled (40%, n = 11). Most samples were collected at the Faro district (n = 11), while the remaining were collected at Setúbal (n = 7), Lisboa (n = 6), and Santarém (n = 3) (Figure 1).

After inoculating the 27 samples collected in MacConkey agar, it was possible to obtain 84 isolates presumptively identified as lactose-fermenting Enterobacteriaceae. Only six samples failed to produce *E. coli* typical colonies; as such, it was possible to isolate *E. coli* from 78% (n = 21) of the sampled animals. Data on all the sampled animals are shown in Appendix A.

### 2.2. Characterization of Isolates’ DNA Fingerprint

Fingerprinting profiles were obtained for all 84 isolates, as shown in Figure 2. Based on the average reproducibility value obtained, the cut-off value for clone identification was set as 68.8%. One isolate from each clonal type was randomly selected for further characterization, except in the cases in which the clonal groups included isolates obtained from samples collected from different bird species and locations; in these cases, both isolates were selected for further analysis, resulting in the selection of 33 representative isolates (Figure 3).

### 2.3. Phylogenetic Grouping

More than one-third of the isolates under study belonged to the phylogenetic group D (36%, n = 12), with the remaining being allocated to group B2 (27%, n = 9), group B1 (24%, n = 8), and group A (12%, n = 4) (Figure 4). As such, more than half of the tested isolates belonged to groups D and B2. No significant differences were found between the isolates’ phylogenetic group and any other variable under study. All data on the isolates’ phylogenetic group are available in Appendix A.

### 2.4. Characterization of the Isolates’ Virulence Profiles

Regarding the virulence profiles of the representative *E. coli* isolates under study (n = 33), most isolates presented gelatinase activity (88%, n = 29), almost half were able to produce biofilm (45%, n = 15), and some were able to produce α-hemolysis (18%, n = 6). None of the isolates under study displayed protease, DNase, or lecithinase activities. Of the 15 isolates able to produce biofilm, almost half (47%, n = 7) were classified as strong producers.

The isolates’ Virulence Index values ranged from 0 (n = 1) to 0.5 (n = 1), with an average value of 0.262 (σ = 0.10). The one isolate with a virulence index of 0.5 was collected from a *Falco tinnunculus* individual in Faro, being positive for gelatinase, biofilm, and α-hemolysin production. All data on the virulence profiles of the *E. coli* isolates are shown in Appendix A.

No significant correlation was found between the Virulence Indexes of the isolates and bird species sampled, sample location, or isolates’ phylogenetic group. A tendency to a positive correlation was observed between gelatinase activity and a higher score of biofilm production (*p* = 0.069).

### 2.5. Characterization of Isolates’ Resistance Profiles

As shown in Table 2, piperacillin (42%), tetracycline (33%), and doxycycline (30%) were the antimicrobials associated with higher resistance rates. No isolate displayed resistance to amoxicillin/clavulanic acid. Of the 33 isolates tested, 2 (6%) were classified as multidrug resistant, considering the definition by Magiorakos [17]. The isolates MAR Index values ranged from 0 to 0.667, with an average of 0.168 (σ = 0.20). The two isolates with higher MAR indexes were both obtained from samples collected from birds belonging to the same species, *Parabuteo unicinctus*. According to the modified double disk synergy test, only two isolates (6%) were able to produce ESBLs. All data on the resistance profiles are available in Appendix A.

No significant differences (*p* > 0.05) were found between the isolates’ MAR index and animals’ species of origin, sampling location, and phylogenetic group. There was also no significant statistical difference between the MAR index and biofilm production score, nor between the MAR index and virulence index. A very strong positive correlation between an isolate being resistant to doxycycline and tetracycline was observed, as well as a strong correlation between an isolate being resistant to marbofloxacin and enrofloxacin; moreover, 10 other moderate correlations between the isolates’ resistance to other antibiotics were detected (Figure 5). A moderate correlation was also found between the capacity of an isolate to produce ESBLs and being resistant to cephalexin (Figure 5).

## 3. Discussion

While the potential impact of antimicrobial resistance dissemination in wildlife and companion animals has received more attention over the past years, this study represents, to our knowledge, the first evaluation of the virulence and resistance profiles of bacteria from falconry birds of prey in close contact with both wildlife and humans.

The analysis of the fingerprinting profiles of the 84 isolates tested did not allow us to identify any major cluster; however, 33 clonal types were found, with levels ranging from 1.6 to 99.4. These results reveal a high strain diversity, which is expected given the abundance of environmental variables present in the ecosystems inhabited by these birds. Moreover, it was not possible to draw a comparison between our findings and the other studies, as there are no previous reports available on the characterization of fecal *E. coli* isolates from these captive birds.

As for phylogenetic grouping, it is described that both groups B2 and D primarily include strains with a higher pathogenic potential, while isolates from groups A and B1 are often regarded as commensals [18]. In this study, more than half (63%) of the isolates tested belonged to either group D or B2, revealing the importance of these falconry birds as vectors of *E. coli* strains with pathogenic potential.

Studies on the relationship between the phylogenetic grouping and virulence traits of *E. coli* strains from wild birds are scarce. Still, contrary to a previous study from 2012 [19], in which isolates from the phylogenetic groups B2 and D were associated with a higher number of virulence factors, no significant differences between these parameters were found in our study. This discrepancy may be due to differences in the virulence factors assessed in our study compared to those previously identified as contributing to this correlation, and to the fact that the animal species studied by these researchers were not birds of prey [19]. Also, in contrast to previous studies [13], both isolates from our study that were able to produce ESBLs were not classified in group B2, but in group D.

This work also focused on identifying relationships between the phenotypic expression of virulence factors, allowing us to recognize a positive tendency between an isolate’s ability to present gelatinase activity and having a higher biofilm-producing capacity. Gelatinase activity is involved in tissue damage, while biofilms are bacterial communities associated with chronic infections and treatment failure [20]. As such, biofilm production can often be associated with the expression of gelatinase as some authors consider this enzyme necessary for the establishment of biofilms; however, the association between these two virulence factors has not been fully established [20,21].

As for antibiotic resistance, our results on tetracycline and trimethoprim–sulfamethoxazole resistance are similar to those from a previous study focusing on several Iberian synanthropic birds, which reported resistance levels of 21.3% and of 18.9%, respectively [14]. The authors also described a low incidence of resistance to amoxicillin–clavulanic acid and gentamicin, which agrees with our results. The percentage of ESBL-producing *E. coli* obtained (6%) was also similar to the one reported in a study conducted on synanthropic pigeons from the Lisbon area (9%) [22]. According to the World Health Organization, gentamicin, enrofloxacin, marbofloxacin, and piperacillin are considered highly important antimicrobials for human health [23]. In addition, the World Organization for Animal Health (WOAH, founded as OIE), considers cephalexin as a highly important antimicrobial agent for veterinary medicine [24]. As such, it is important to refer that at least one of the isolates evaluated in our study presented resistance to these relevant antimicrobials.

On the other side, other published studies on wild birds of prey from Portugal described higher levels of resistance to tetracycline (75%), amoxicillin/clavulanic acid (38.9%), and gentamicin (19.4%) [25]. As such, the resistance profiles of the *E. coli* isolates from falconry birds evaluated in our study are more similar to the profiles of isolates from synanthropic species than to those from their wild counterparts, or even from other pets. In fact, a study conducted on 265 companion birds concluded that any of the cloacal swabs collected presented isolates with the ability to produce ESBL [26].

Another factor that must be considered when comparing resistance profiles of bacteria from wild and falconry birds is that, besides being in contact with synanthropic birds, these birds of prey also interact with humans, who can also act as disseminators of resistance determinants. Although the information available on this subject is scarce, one study has already revealed that the ingestion of the raw food provided by the handlers may allow the direct transmission of multidrug-resistant bacteria to these birds [27].

The very strong correlation between the isolates’ resistance to tetracycline and doxycycline (0.99) and the strong correlation between their resistance to marbofloxacin and enrofloxacin (0.66) identified in this study were expected as these compounds belong to the same antimicrobial categories. The association between the capacity of an isolate to produce ESBLs and being resistant to cephalexin can be explained by the fact that ESBLs are β-lactamases capable of conferring resistance to cephalosporins [28].

The MAR index and V. Index allow us to assess the threat level of bacterial isolates. These can be classified as low threat when their virulence index is below the cut-off value, but their MAR index is above it, and as high threat when their virulence and MAR indexes are higher than or equal to their cut-off values [29]. Considering the MAR index and V. Index values obtained in this study, six of the tested isolates can be classified as low-threat isolates, and one isolate can be classified as a high-threat isolate.

Ultimately, the bacteria present in the feces of captive birds of prey can represent a risk, as they may express virulence factors and act as disseminators of antimicrobial resistance determinants. This might be of great relevance under the One Health approach, as these animals may potentially contribute to the spread of potentially zoonotic agents from wildlife bacterial reservoirs to humans, other animals, and the environment. Further characterization of bacterial strains from captive birds of prey should be performed to understand the potential public health risk associated with these animals, aiming to improve management and housing practices in order to minimize both the acquisition and dissemination of virulent and resistant bacterial strains.

## 4. Materials and Methods

### 4.1. Sample Collection

From December 2022 to January 2023, cloacal samples from 27 birds inhabiting four Portuguese districts (Lisboa, Santarém, Setúbal, and Faro) (Figure 1) were collected during routine animal management procedures by trained personnel. The animals selected for the study were captive-born birds of prey simultaneously used for avifauna control and as demonstration birds in animal exhibitions. All the Portuguese companies working in these fields, with animals fitting these conditions, were contacted for sample collection. Individuals in reproductive programs, birds whose training had not yet been finished, and animals who were not working at the time of the study were not selected for sample collection.

While the animal was focused on a small piece of food strategically placed on the glove of the keeper, a sterile AMIES swab (VWR, Brescia, Italy) was inserted into the cloaca, by gently rotating the tip against the mucosa. After sample collection, swabs were kept at 4 °C and transported to the Laboratory of Microbiology and Immunology of the Faculty of Veterinary Medicine, University of Lisbon, for further processing.

### 4.2. Bacterial Isolation and Identification

Swabs were inoculated in Brain Heart Infusion broth and incubated for 48 h at 37 °C [30], after which bacterial suspensions were spread onto MacConkey agar (VWR, Leuven, Belgium) surface, and incubated at 37 °C for 48 h. After incubation, four lactose-fermenting colonies, each surrounded by a halo of bile salt precipitation, were selected for further identification [31]. Isolates were tested using oxidase reagent swabs (VWR, Leuven, Belgium) and characterized by Gram staining. The identification of presumptive *E. coli* isolates (Gram-negative and oxidase-negative bacilli) was confirmed using the Indole, Methyl Red, Voges–Proskauer, and Citrate test (IMViC test). Typically, an isolate that is positive for both the Indole and Methyl Red tests, and negative for hydrogen sulfide production, the Voges–Proskauer, and citrate utilization tests can be identified as *E. coli* [32]. The selected lactose-fermenting colonies formed by Gram-negative and oxidase-negative bacilli that did not present the typical IMViC profile were submitted to biochemical identification using API20E (bioMérieux, Marcy-l’Etoile, France) galleries, which was carried out according to the instructions provided by the manufacturer, allowing us to confirm the isolates’ identities as *E. coli*.

### 4.3. Bacterial DNA Extraction

Each *E. coli* isolate was submitted to an overnight incubation at 37 °C in Brain Heart Infusion Agar. Then, approximately 10 µL of the bacterial cultures formed by the isolates under testing, and by both *E. coli* J96 and *E. coli* KS52 reference strains, were collected using a loop, and resuspended in 100 μL of tris-ethylenediaminetetraacetic acid buffer (VWR, Philadelphia, PA, USA) supplemented with 0.1% Tween 20 (Fisher Bioreagents, Pittsburgh, MA, USA). As described by Dashti’s team [33], bacterial suspensions were then incubated at 100 °C for 10 min, after which they were centrifuged at 14,000 rpm for another 10 min. The resulting supernatant was collected, and the DNA concentration and purity were assessed using a NanoDrop™ Spectrophotometer (ThermoFisher Scientific, Waltham, MA, USA). DNA purity was determined based on the 280/260 absorbance ratio, which should be of approximately 1.8 for the DNA suspension to be considered pure. Subsequently, all DNA suspensions were diluted in sterile and ultrapure water (VWR, Philadelphia, PA, USA) until they reached a standardized concentration of 50 ng/μL and were subsequently kept at −20 °C until further use.

### 4.4. DNA Fingerprinting

All DNA samples were submitted for DNA fingerprinting analysis, using an enterobacterial repetitive intergenic consensus (ERIC) polymerase chain reaction (PCR) protocol [34]. PCR mixtures were performed in a final volume of 25 μL, consisting of 2 μM of ERIC2 primer (5′-AAGTAAGTGACTGGGGTGAGCG-3′) (STABVIDA, Caparica, Portugal), 100 ng DNA, 10 μL PCR-grade water (VWR, Philadelphia, PA, USA), and 12.5 μL of MasterMix (NZYtaq 2x Green) (NZYtech, Lisbon, Portugal). In the negative control reaction, the volume corresponding to DNA was replaced with water.

The amplification protocol included an initial denaturation for 7 min at 95 °C, followed by 30 cycles of denaturation at 90 °C for 30 s, annealing at 52 °C for 1 min, and extension at 72 °C for 8 min. A final extension step was then performed at 72 °C for 16 min [34]. Then, 10 μL of the amplification products were revealed through agarose gel electrophoresis with 1.5% agarose (NZYtech, Lisbon, Portugal) in 0.5X Tris–Borate–EDTA buffer (AppliChem, Darmstadt, Germany) and stained with Green Safe (NZYtech, Lisbon, Portugal). NZYDNA Ladder III (NZYtech, Lisbon, Portugal) was used as a molecular weight marker, and the electrophoresis ran for 2 h at 70 V. The gels were visualized using a UV light transilluminator, and the images were processed and recorded using the Bio-Rad ChemiDoc XRS imaging system (Bio-Rad Laboratories, Hercules, CA, USA).

The fingerprinting profiles of all isolates were compared using BioNumerics 6.6 (Applied Maths, Kortrijk, Belgium), with a hierarchical numerical process based on the Pearson correlation coefficient and the unweighted pair group method with arithmetic average as the agglomerative clustering. The reproducibility value was determined as the average value for seven pairs of duplicates. Finally, one representative isolate from each clonal type was randomly selected for further characterization.

### 4.5. Phylogenetic Grouping

A quadruplex PCR was used to determine the phylogenetic grouping of the selected isolates by evaluating the presence of the genes *gadA*, *chuA*, and *yjaA*, in addition to the DNA fragment TSPE4.C2 [35]. For a reaction volume of 20 μL, the mixture consisted of the primers represented in Table 3 (STABVIDA, Caparica, Portugal), each at a final concentration of 1 μM, 50 ng DNA, 5.8 μL of PCR-grade water, and 10 μL of MasterMix (NZYtaq 2× Green). In the negative control, water was added instead of DNA. DNA from the isolates belonging to three different phylogenetic groups, *E. coli* J96 (group B2), *E. coli* KS52 (group A), and isolate 16.1 (group D), were also tested as positive controls. Isolate 16.1 was used as a positive control for the detection of the TSPE4.C2 fragment, after the resulting PCR products were characterized by Sanger sequencing (STABVIDA, Caparica, Portugal), to determine their expected length and sequence.

The amplification protocol included an initial denaturation for 4 min at 94 °C, followed by 30 cycles of denaturation at 94 °C for 30 s, annealing at 65 °C for 30 s, and extension at 72 °C for 30 s, ending with a final extension step at 72 °C for 5 min [35,36]. The PCR products were revealed through electrophoresis, using a gel with 2% agarose in 0.5X Tris–Borate–EDTA buffer, stained with Green Safe. Ladder VI (NZYtech, Lisbon, Portugal) was used as a molecular weight marker, and the electrophoresis was run at 70 V for 2 h. The gels were visualized using the hardware and software previously described.

The data allowed us to determine the isolates’ phylogenetic group, as shown in Table 4. Randomly selected isolates were used to perform 10% replicas.

### 4.6. Isolates’ Virulence Profiles

The isolates’ virulence profiles were assessed using specific media [29,31]. Results were read after incubation at 37 °C for 24 to 48 h, except for gelatinase activity. A 10% replica was evaluated in each test.

Hemolysin expression was tested after inoculation of each isolate in Columbia Agar with 5% sheep blood (bioMérieux, Marcy-l’Etoile, France), followed by an incubation for detecting both β-hemolytic and α-hemolytic bacteria [31].

DNase production was assessed with a DNase medium (Thermo Scientific Remel, Lenexa, KS, USA) supplemented with 0.01% toluidine blue (Merck, Darmstadt, Germany), using *Staphylococcus aureus* ATCC 25923 and *E. coli* ATCC 25922 as positive and negative controls, respectively [31].

Lecithinase activity was determined using Tryptic Soy Agar (VWR, Leuven, Belgium), supplemented with 10% egg yolk emulsion (VWR, Leuven, Belgium), using *Pseudomonas aeruginosa* ATCC 27853 and *E. coli* ATCC 25922 as the positive and negative controls [31].

Protease activity was evaluated using Skim Milk (VWR, Leuven, Belgium) Agar, with *P. aeruginosa* ATCC 27853 and *S. aureus* ATCC 29213 as the positive and negative controls [31].

Gelatinase activity was detected using Nutrient Gelatin Agar (Oxoid, Hampshire, UK), with *P. aeruginosa* Z25.1 and *E. coli* ATCC 25922 as the positive and negative controls. Gelatinase-positive isolates promoted the liquefaction of the medium, after a first incubation at 37 °C for 24 h followed by a second incubation for 45 min at 4 °C [31].

Biofilm production was assessed using Red Congo Agar, composed of Brain Heart Infusion Broth (VWR, Leuven, Belgium) at 3.7%, Bacteriological Agar (VWR, Leuven, Belgium) at 1.4%, Sucrose (Sigma^®^, Steinheim, Germany) at 5%, and Red Congo reagent (Sigma-Aldrich, Steinheim, Germany) at 0.08%, and using *P. aeruginosa* ATCC 27853 and *E. coli* ATCC 25922 as the positive and negative controls [31].

Finally, the virulence index of each isolate was determined as the quotient between the number of positive virulence factors expressed by an isolate and the number of virulence factors tested [29,31].

### 4.7. Isolates’ Antibiotic Resistance Profiles

Isolates’ antimicrobial resistance profile against nine antibiotics from five different classes, was determined using the disk diffusion method, according to the established guidelines [37,38,39]. The antibiotics tested included tetracycline (TE, 30 μg), gentamicin (CN, 10 μg), sulfamethoxazole/trimethoprim (SXT, 25 μg), piperacillin (PRL, 100 μg), amoxicillin/clavulanic acid 2:1 (AMC, 30 μg), doxycycline (DO, 30 μg), enrofloxacin (ENR, 5 μg), cephalexin (CL, 30 μg) (Oxoid, Hants, United Kingdom), and marbofloxacin (MAR, 5 μg) (Liofilchem, Roseto, Italy). The reference strains, *E. coli* ATCC 25922 and *S. aureus* ATCC 25923, were also tested as controls. Moreover, the isolates were randomly selected to perform a 10% replica.

The multiple antimicrobial resistance index (MAR index) of each tested isolate was determined as the quotient between the number of antimicrobials to which the isolate was resistant and the total number of antimicrobials tested [29,31]. Isolates were also categorized as multidrug-resistant (MDR), extensively drug-resistant (XDR), or pandrug-resistant (PDR), considering their resistance profile towards distinct antibiotic classes [17].

Lastly, for detecting the isolates’ ability to produce extended-spectrum beta-lactamases (ESBLs), the modified double disk synergy test was applied [40]. A bacterial suspension of 10^8^ CFU/mL in 0.9% saline solution (VWR, Leuven, Belgium) was spread onto the surface of Mueller–Hinton agar plates (Oxoid, Hants, UK). The three antimicrobial disks were then sequentially placed on the surface of the agar while being kept at a distance of 20 mm, as follows: ceftazidime (CAZ, 30 μg) and cefotaxime (CTX, 30 μg) (Oxoid, Hants, UK) were placed in each end, and amoxicillin/clavulanic acid (AMC, 30 μg) in the middle. The agar plates were then incubated at 37 °C for 16 to 18 h, after which the isolates were classified as ESBL-producers if the inhibition halo surrounding any of the two cephalosporines showed a clear-cut increase towards the AMC disk [41].

### 4.8. Data Analysis

Microsoft Excel (Microsoft Corporation, Redmond, WA, USA) was used to determine the means and frequencies, and further data analysis was performed using the SAS software version 9.4 (SAS Institute Inc., Cary, NC, USA).

Cumulative logistic regression models with the logit link function for ordinal response variables (PROC LOGISTIC) were used to test the association between bird species, *E. coli* phylogenetic group, sampling location, the isolates’ virulence index (V. index) and MAR index, and biofilm-forming ability. In these models, the probability (odds ratio) of having a higher virulence index, MAR index, and biofilm-forming ability was modeled. Final models were obtained through manual backward elimination at a threshold of *p* ≤ 0.05. The Spearman’s coefficient was used to study the relationship between the MAR and virulence indexes, and between the level of antimicrobial resistance to each antibiotic tested. The Kruskal–Wallis test was applied to test the effect of gelatinase activity on biofilm production. Differences were considered significant when *p* ≤ 0.05, whereas a tendency was defined as 0.05 < *p* ≤ 0.10.

## 5. Conclusions

Despite the increasing concern about virulent and antibiotic-resistant bacteria and their implications on human health, there seems to be a lack of information on the presence of these strains in other ecosystems. Falconry intrinsically connects the three pillars of the One Health concept, representing a perfect model to study the effects of these interactions. Our results confirm the importance of captive birds of prey as reservoirs of virulent and resistant bacteria, stressing the need for more studies on their role as a relevant link between humans and wildlife.

## Figures and Tables

**Figure 1 antibiotics-13-00379-f001:**
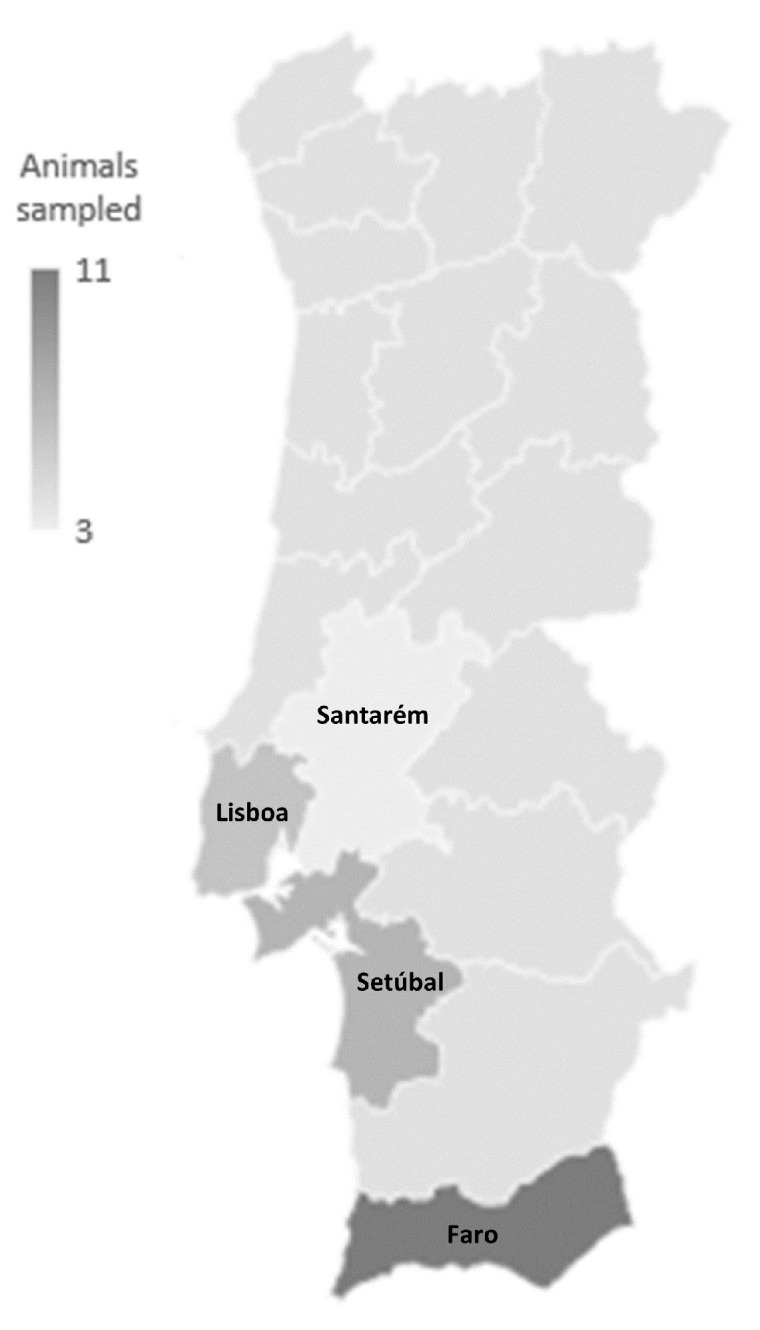
Map of sampling locations in mainland Portugal.

**Figure 2 antibiotics-13-00379-f002:**
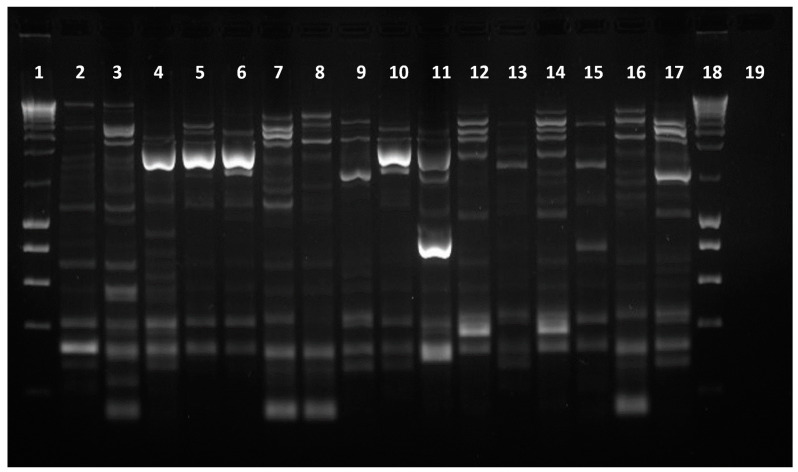
ERIC-PCR fingerprinting profiles of 16 of the *E. coli* isolates under study. Lanes 1 and 18—ladder III (NZYtech, Lisbon, Portugal); lanes 2 to 17—isolates under study; lane 19—negative control. The original gel can be found in Appendix A.

**Figure 3 antibiotics-13-00379-f003:**
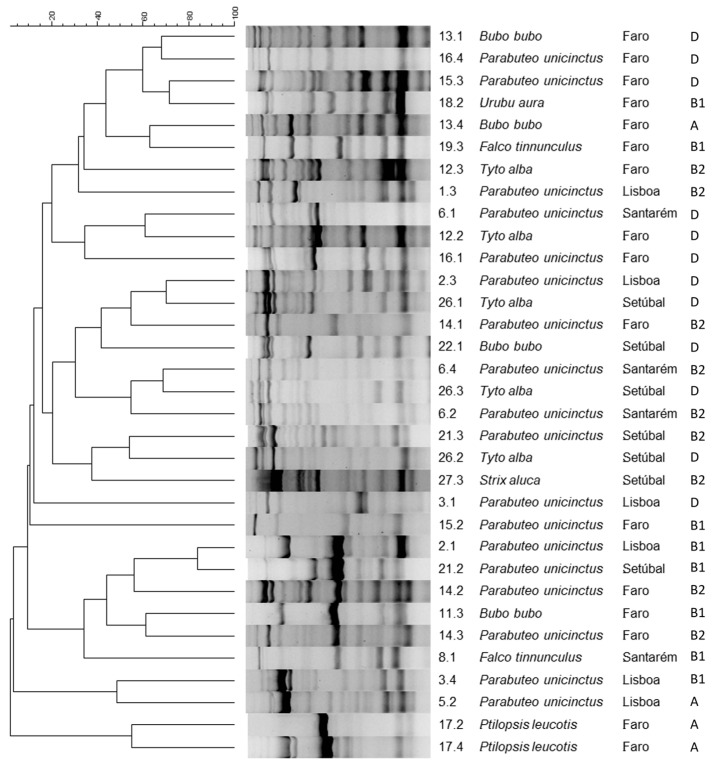
Dendrogram of the 33 representative isolates selected based on the fingerprinting profiles obtained by ERIC-PCR, built using BioNumerics 6.6. Includes information on bacterial isolates’ identification number, animal species of origin, location of the animal sampled, and isolates’ phylogenetic group.

**Figure 4 antibiotics-13-00379-f004:**
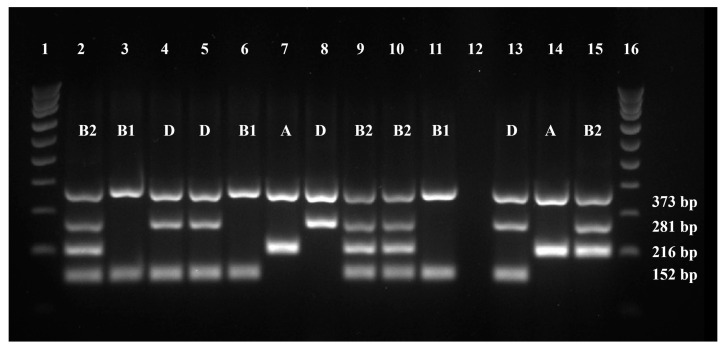
Quadruplex PCR profiles of 10 *E. coli* isolates and respective phylogenetic groups. Lanes 1 and 16—ladder VI (NZYtech, Portugal); lanes 2 to 11—isolates under study; lane 12—negative control, lanes 13 to 15—positive controls. The original gel can be found in Appendix A.

**Figure 5 antibiotics-13-00379-f005:**
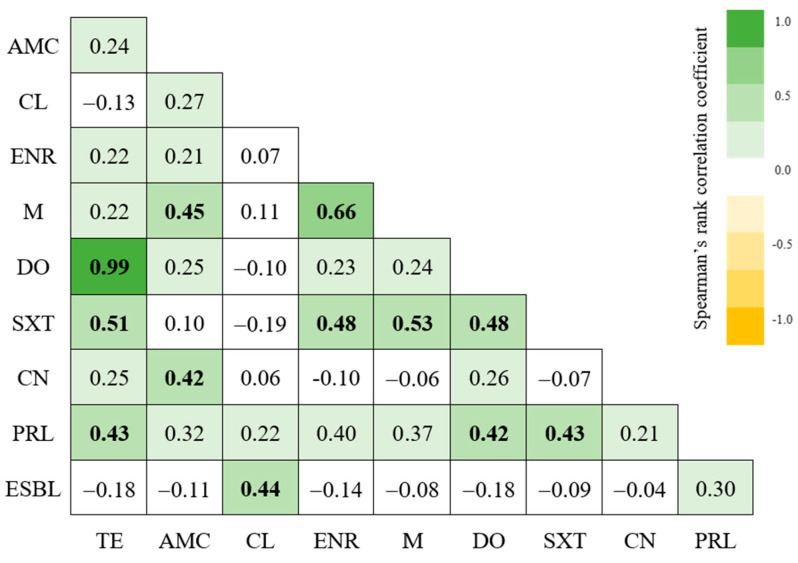
Spearman’s rank correlation of resistance to tested antibiotics and ESBL production. Statistically significant results are shown in bold (*p* > 0.05). Tetracycline (TE); Gentamicin (CN); Amoxicillin/Clavulanic Acid (AMC), Cephalexin (CL), Enrofloxacin (ENR), Trimethoprim/Sulfamethoxazole (SXT); Piperacillin (PRL); Doxycycline (DO); Marbofloxacin (M); Extended-Spectrum Beta-Lactamase (ESBL).

**Table 1 antibiotics-13-00379-t001:** Number of swab samples collected from each bird species (n = 27).

Species	Common Name	Samples (n = 27)
*Parabuteo unicinctus*	Harris’s hawk	11
*Bubo bubo*	Eurasian Eagle-Owl	4
*Falco tinnunculus*	Common Kestrel	4
*Tyto alba*	Western Barn Owl	2
*Bubo bengalensis*	Rock Eagle-Owl	1
*Bubo virginianus*	Great Horned Owl	1
*Falco biarmicus*	Lanner Falcon	1
*Ptilopsis leucotis*	Northern White-faced Owl	1
*Strix aluco*	Tawny Owl	1
*Urubu aura*	Turkey vulture	1
Total		27

**Table 2 antibiotics-13-00379-t002:** Antimicrobial susceptibility profile of isolates under study (n = 33). Not found (-).

Antimicrobial Category	Antimicrobial Agent	Disk Content	Number of Isolates (n = 33)		
			Susceptible	Intermediate	Resistant
Tetracyclines	Tetracycline	30 μg	22	-	11
	Doxycycline	30 μg	22	1	10
Aminoglycosides	Gentamicin	10 μg	32	-	1
Folate pathway inhibitors	Sulfamethoxazole/ Trimethoprim	25 μg	29	-	4
Penicillins + β-lactamase inhibitors	Amoxycillin/ Clavulanic Acid	30 μg	28	5	-
Antipseudomonal Penicillins	Piperacillin	100 μg	19	-	14
First-generation Cephalosporin	Cephalexin	30 μg	10	19	4
Fluoroquinolone	Enrofloxacin	5 μg	25	5	3
	Marbofloxacin	5 μg	30	-	3

**Table 3 antibiotics-13-00379-t003:** Primer sequences (5′-3′) used for determining the *E. coli* phylogenetic groups [35].

Marker	Primer Direction	Primer Sequence (5′-3′)	Product Length (bp)	Positive Control
*gadA*	Forward	GATGAAATGGCGTTGGCGCAAG	373	*E. coli* J96 *E. coli* KS52 Isolate 16.1
	Reverse	GGCGGAAGTCCCAGACGATATCC		
*chuA*	Forward	ATGATCATCGCGGCGTGCTG	281	*E. coli* J96 Isolate 16.1
	Reverse	AAACGCGCTCGCGCCTAAT		
*yjaA*	Forward	TGTTCGCGATCTTGAAAGCAAACGT	216	*E. coli* J96 *E. coli* KS52
	Reverse	ACCTGTGACAAACCGCCCTCA		
TSPE4.C2	Forward	GCGGGTGAGACAGAAACGCG	152	Isolate 16.1
	Reverse	TTGTCGTGAGTTGCGAACCCG		

**Table 4 antibiotics-13-00379-t004:** Identification key for phylogenetic grouping of the *E. coli* isolates [35].

Markers	Product Length (bp)	Phylogenetic Group						
		A		B1	B2		D	
*gadA*	373	+	+	+	+	+	+	+
*chuA*	281				+	+	+	+
*yjaA*	216		+		+	+		
TSPE4.C2	152			+		+		+

Legend: + fragment presence.

## Data Availability

The data presented in this study are available in the article and Appendix A. More details can be provided upon reasonable request to the correspondence contacts.

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
