# Peer review of "First Insights on Resistance and Virulence Potential of Escherichia coli from Captive Birds of Prey in Portugal"

_antibiotics, 2024, doi:10.3390/antibiotics13050379_

Round 1
Reviewer 1 Report
Comments and Suggestions for Authors
1. Study location in Portugal should be reflected in the title.
2. There are general issues of grammar, spellings, punctuations throughout the manuscript.
3. Virulence and resistance traits are borne by bacterial isolates and not the birds.
4. The common names of the avian species in Table 1 should be indicated and the title should reflect avian species and not animals.
5. The organs cultured and how the outcome of API 20E was interpreted should be stated.
6. The title of all tables and figures should be clear and separated from the legends and explanatory footnotes.
7. How were the antibiotics selected and why were 3rd and 4th generation cephalosporins not included?
Comments on the Quality of English Language
There is serious need to improve on the quality of grammar throughout the manuscript, as pointed out above in comments to the authors. Ideally, a scientist fluent in spoken and written communication in English should be engaged for this purpose.
Author Response
Dear Review,
We would like to thank you for the comments and the discussion points presented. We agree that they will improve the quality and understanding of the manuscript. Please see below the response to each of your question. Also, all changes performed in the revised version of the manuscript are presented highlighted in yellow.
“1. Study location in Portugal should be reflected in the title.”
- We agree with your suggestion and the reference to Portugal was added to the title of the manuscript (line 3).
“2. There are general issues of grammar, spelling, punctuations throughout the manuscript.”
- We understand your comment. The manuscript was revised and several corrections were made to address errors in grammar, punctuation, and sentence structure, as presented highlighted in yellow over the manuscript text.
“3. Virulence and resistance traits are borne by bacterial isolates and not the birds.”
- We understand your remark. We have clarified this point, as you may see in line 79.
“4. The common names of the avian species in Table 1 should be indicated and the title should reflect avian species and not animals.”
- We agree with your comment. Table 1 was revised in accordance as well as the title, as you may see in table 1, line 97.
“5. The organs cultured and how the outcome of API 20E was interpreted should be stated.”
- We thank you for your comment. Reference to this was added to the revised manuscript in lines 302, 305-306.
“6. The title of all tables and figures should be clear and separated from the legends and explanatory footnotes.”
- We thank you for this comment. We have improved the title of the figures and separated the legends and footnotes as suggested. Changes can be observed in lines 120-124; 126-129;140-143.
“7. How were the antibiotics selected and why were 3rd and 4th generation cephalosporins not included?”
- We thank you for this comment. The antibiotics tested were chosen from a One Health perspective, trying to provide answers relevant to all fields of this sector. As such, antibiotics tested in this study included compounds important not only to veterinary medicine but also to public health. Considering that, as 3rd and 4th generation cephalosporins are not commonly used in birds, they were not a focus on our study. However, when testing ESBL production via the modified double disk synergy test we also studied two 3rd generation cephalosporins, namely ceftazidime and cefotaxime, as presented on manuscript in line 422-423.
Best Regards
Reviewer 2 Report
Comments and Suggestions for Authors
Your study presents a comprehensive exploration of the role played by the enteric microbiota of falconry birds of prey, with a particular focus on Escherichia coli, as indicators of resistance and virulence dispersion in humans. The text is well-written, providing all necessary information for understanding the research's objectives and implications. Your emphasis on the study's significance in elucidating the potential risks associated with virulence factors and antimicrobial resistance dissemination from captive birds of prey is particularly noteworthy.
However, the small sample size raises concerns about the representativeness of the findings. Additionally, while your decision to select distinct colonies per sample is acknowledged, it would be beneficial to clarify whether there were distinguishing characteristics in the MacConkey agar that guided this selection process. In this selection there is a possibility of analyzing the exact same strain (in each sample) so the 84 isolates tested is not a representative number of isolates. It is important to address whether potentially identical strains were excluded from the total number of isolates (84) tested to ensure the robustness of the sample.
Author Response
Dear Reviewer,
We would like to thank you for the comments and the discussion points presented. The sample size was influenced by the inclusion criteria defined for the study - birds of prey who work in avifauna control and also as display birds, born in captivity, without known health problems and not participating in any ongoing reproductive programs. Unfortunately, samples from these animals are not as easily available as other pets such as cats and dogs, and so despite contacting all Portuguese companies working in this field, only 27 birds met all inclusion criteria (please see lines 284-292).
Considering the methodology applied, as showed in sections 2.2 and 3.1, the 84 isolates were obtained by selecting 4 lactose fermenting colonies from cultures originating from each sampled bird. However, all isolates were characterized through fingerprinting analysis, and only one representative isolate from each clonal type was included in the study, making them relevant as they represent a different strain, excluding identical strains even if originated from the same bird (please see lines 352-353).
Best regards.